# Implications of Temperate Agroforestry on Sheep and Cattle Productivity, Environmental Impacts and Enterprise Economics. A Systematic Evidence Map

**Matthew W. Jordon** [1,*] **, Kathy J. Willis** [1] **, William J. Harvey** [2,3] **, Leo Petrokofsky** [2] **and Gillian Petrokofsky** [1,2]

1 Department of Zoology, University of Oxford, Mansfield Road, Oxford OX1 3SZ, UK; kathy.willis@zoo.ox.ac.uk (K.J.W.); gillian.petrokofsky@zoo.ox.ac.uk (G.P.)
2 Oxford Systematic Reviews, Unit 132, 266 Banbury Road, Oxford OX2 7DL, UK; william.harvey@arch.ox.ac.uk (W.J.H.); leo.petrokofsky@oxsrev.org (L.P.)
3 School of Archaeology, University of Oxford, 36 Beaumont Street, Oxford OX1 2PG, UK
* Correspondence: matthew.jordon@zoo.ox.ac.uk

**Abstract:** The environmental impacts of ruminant livestock farming need to be mitigated to improve the sustainability of food production. These negative impacts have been compounded by the increased spatial and cultural separation of farming and forestry across multiple temperate landscapes and contexts over recent centuries, and could at least in part be alleviated by re-integration of livestock and trees via agroforestry systems. Such integration also has the potential to benefit the productivity and economics of livestock farming. However, the delivery of hoped-for benefits is highly likely to depend on context, which will necessitate the consideration of local synergies and trade-offs. Evaluating the extensive body of research on the synergies and trade-offs between agroforestry and environmental, productivity and economic indicators would provide a resource to support context-specific decision making by land managers. Here, we present a systematic evidence map of academic and grey literature to address the question "What are the impacts of temperate agroforestry systems on sheep and cattle productivity, environmental impacts and farm economic viability?". We followed good practice guidance from the Collaboration for Environmental Evidence to find and select relevant studies to create an interactive systematic map. We identified 289 relevant studies from 22 countries across temperate regions of North and South America, Australasia and Europe. Our preliminary synthesis indicates that there is an emerging evidence base to demonstrate that temperate agroforestry can deliver environmental and economic benefits compared with pasture without trees. However, to date measures of livestock productivity (particularly weather-related mortality and heat- and cold-stress) have received insufficient attention in many temperate agroforestry systems. The evidence base assembled through this work provides a freely accessible resource applicable across temperate regions to support context-specific decision making.

**Keywords:** systematic map; temperate agroforestry; silvopasture; sheep; cattle; ruminants

## 1. Introduction

Management of agricultural land has an important role to play in addressing the twin challenges of climate change and biodiversity loss [1,2]. In particular, the environmental impacts of ruminant livestock farming need to be mitigated to improve the sustainability of food production, amid increasing demand for animal-sourced foods driven by human population growth and rising affluence [3,4]. Currently, domestic sheep and cattle are a significant source of anthropogenic greenhouse gas emissions

(GHGEs) [5], and can contribute to air pollution [6], increased flood hazard [7], reduced water quality and enhanced soil erosion [8]. These negative environmental impacts have, in temperate regions, been compounded by the increased spatial and cultural separation of farming and forestry across multiple landscapes and contexts over recent centuries, along with more recent widespread removal of trees and other woody landscape elements as part of agricultural intensification [9–12]. Re-integration of trees and farming systems via a diverse suite of practices collectively termed agroforestry [13], has the potential to at least in part alleviate these environmental harms (Table 1, and citations therein).

**Table 1.** Selected environmental impacts of ruminant livestock production, used as outcomes in the systematic map, and how trees could potentially mitigate these impacts. Natural capital assets and societal benefits affected [14] are also indicated.

| Environmental Indicators | Ruminant Action | Details of Negative Environmental Impact Caused by Ruminants | Ecosystem Services That Can Be Provided by Trees to Mitigate This Impact | Natural Capital Assets Affected | Societal Outcome Affected |
|---|---|---|---|---|---|
| Greenhouse gas emissions and carbon | Eructation + urination and defecation (i.e., manure production) | Emissions: Methane from eructation and manure Nitrous oxide from manure and fertiliser applications Carbon dioxide from machinery and embedded in animal feed production [5] | Carbon sequestration in above- and below-ground tree biomass and soil [15,16] | Air and soil | Stable climate |
| Air quality | | Emissions of air pollutants, e.g., ammonia [6] | Particulate capture by tree leaves [17] | Air | Clean air |
| Water quality | | Nutrient loss in run-off from fields into groundwater and watercourses [8] | Nutrient capture by tree roots [18] | Water | Clean water |
| Water quantity | Trampling and grazing pressure | Reduced water infiltration caused by soil compaction leading to increased water runoff from fields [19] | Increased water infiltration into soil facilitated by tree roots [20,21] and increased transpiration rate of trees [22] | Soil and water | Flood hazard protection |
| Soil erosion | | Soil erosion [8] | Slope stabilisation and sediment capture by tree roots [23] | | Sustained basis for food production |

Simultaneously, increasing tree cover on a global scale is attracting international policy and academic attention as a so-called nature-based solution to contribute to climate change mitigation and biodiversity protection [24–26]. Managed grasslands grazed by domestic ruminants are often identified as suitable locations for afforestation and some simplistically argue that ruminant livestock production should be wholly replaced by forest restoration, e.g. [27]. However, integrating instead of simply replacing livestock with trees, i.e., agroforestry, is widely recognised as having a role to play in increasing tree cover for environmental reasons whilst enabling continued food production [1,24,28,29]. Furthermore, temperate agroforestry can improve farm financial viability [30,31] and deliver livestock productivity and welfare benefits such as increased pasture production [32] and reduced livestock heat and cold stress [33–35].

Although agroforestry systems as a concept were developed in the tropics, there is now a substantial body of evidence from temperate regions on the environmental, productivity and economic implications of integrating trees and food production [13]. Previous temperate evidence syntheses have focused on a limited geographic area, e.g., [36–38], or single productivity or environmental indicators, e.g., [39–42]. There is yet to be a systematic compilation of agroforestry evidence across environmental, productivity and economic indicators and temperate regions. This is important because although policymakers may promote agroforestry for environmental reasons, enterprise productivity and economic factors are likely to be key in stimulating land manager uptake [31]. As such, it is key to understand potential synergies and trade-offs in the delivery of hoped-for benefits from agroforestry and how these depend on context, including climatic and site conditions, livestock and tree species, relevant commodity prices and method of integration.

Here, we systematically map the evidence of the impacts of integrating trees (and shrubs/perennial bioenergy crops, i.e., woody vegetation) into temperate ruminant production systems. Systematic map methods provide a rigorous, objective and transparent means of creating a searchable database of relevant academic and grey literature [43], whilst providing an opportunity to characterise the evidence base and highlight knowledge gaps. Our systematic map addressed the question "What are the impacts of temperate agroforestry systems on sheep and cattle productivity, environmental impacts and farm economic viability?" The question was structured following the Population, Intervention, Comparator, Outcome, Location (PICOL) format (Table 2). We focused on studies that had addressed the following environmental impacts: GHGEs, reduced water and air quality, increased flood hazard and enhanced soil erosion (Table 1). A key aim was to establish the published evidence base demonstrating the potential of agroforestry to mitigate these environmental impacts of ruminant production. We also aimed to capture studies that had demonstrated the delivery of these ecosystem services by the tree component of agroforestry systems without specifically considering any mitigation of the livestock component. The database of studies assembled provides a resource to support context-specific decision making by land managers, academics and policymakers across temperate regions, whilst identifying future field-based research priorities and enabling further quantitative meta-analysis.

**Table 2.** Question breakdown of systematic evidence map, following the Population, Intervention, Comparator, Outcome, Location (PICOL) format.

| PICOL Element | Question Element | Details |
|---|---|---|
| Population | Sheep | *Ovis aries* |
| | Beef and dairy cattle | *Bos taurus* |
| Intervention | Agroforestry | Systems with woody perennials, pasture and livestock. This includes silvopasture, shelterbelts, windbreaks, riparian strips, hedges, *dehesa*, *montado*, wood pasture, forest grazing, orchards, woody biofuel and farm woodlands |
| Comparator | | Livestock farming systems with pasture but no trees/shrubs OR forestry systems with trees but no livestock |
| Outcome | Productivity | Understory/pasture herbage productivity |
| | | Livestock mortality |
| | | Livestock growth rate |
| | | Livestock heat stress |
| | | Livestock cold stress |
| | Environmental indicators | Greenhouse gas emissions or carbon stocks/sequestration |
| | | Water quantity |
| | | Water quality |
| | | Air quality |
| | | Soil erosion |
| | Enterprise economics | Financial implications for land manager |
| Location | Temperate systems | Temperate regions of North and South America, Europe and Australasia |

## 2. Materials and Methods

We followed the Collaboration for Environmental Evidence (CEE) guidelines [44] and methodology therein to create our systematic evidence map, and followed the RepOrting standards for Systematic Evidence Syntheses (ROSES) forms to describe our outcomes [45]. Agroforestry networks in English-speaking countries with temperate agroforestry systems, identified from Gordon, Newman and Coleman [13], were contacted for feedback on the suitability of the review questions. A request was also made to these stakeholders for submission of any relevant grey literature that they were

aware of and that might not have been retrieved through bibliographic databases or citation indexes (Appendix A.2).

Details of the search strategy used are given in Table 3 and Appendix A. The first search of bibliographic databases took place on 27 September 2019. The comprehensiveness of the search was estimated using a test list of 23 articles (Appendix A.4). Nine of the 23 test articles were missed by the initial search, so additional search terms were added until all test articles were recovered during the search. The definitive search with results taken forward for screening took place on 2 October 2019. All databases were searched for "All years", i.e., there was no restriction on publication date during the search or screening.

**Table 3.** Methods for each component of the systematic literature search strategy.

| | |
|---|---|
| Search string | For the full search string used, see Appendix A.1. The search string was structured by PICOL elements (Table 2). Terms are joined by "OR" Boolean operators within PICOL elements and "AND" operators between elements. The same search string was used across all bibliographic databases and citation indexes |
| Languages—bibliographic databases | English only |
| Languages—grey literature | English only |
| Bibliographic databases and citation indexes | Web of Science (databases searched listed in Appendix A.1.1), CAB Abstracts, Scopus |
| Organisational websites | Five stakeholder organisations were contacted by email (Appendix A.2) and 15 organisational websites were searched (Appendix A.3) |
| Estimating the comprehensiveness of the search | A test list of 23 articles was compiled (Appendix A.4) from reviews read at the initial scoping stage [13,46,47] |

Only studies that exactly met our PICOL elements (Table 2) were included in the evidence map. For details of the inclusion/exclusion criteria applied to records during screening, see Appendix B.1. All title and abstract screening was carried out in Endnote X9.3. It was not possible to check consistency of screening decisions at these stages due to only one reviewer screening. However, a conservative screening approach was adopted whereby articles were only excluded if they were deemed highly likely to be irrelevant. During title screening, any additional duplicates that had not been detected by Endnote's automated duplicate finder (e.g., due to author or title names being in different formats) were excluded. Records with no title were screened by abstract and included/excluded accordingly. During abstract screening, 799 records with no abstract were identified (Figure 1). These were later rescreened based on title and separated into two groups: higher priority (61 records most likely to be relevant at full text) and lower priority (738 records unlikely to be relevant at full text). Full texts for the 61 higher priority records were searched for and coded accordingly at the full text screening and coding stage.

Article full texts were screened for relevance in the order Location, Population, Intervention, Comparator, Outcome, such that the exclusion reason given is the first of these criteria that the article did not meet. Article full texts were initially screened in batches of 10 by two reviewers, with a Cohen's unweighted kappa calculated for each batch to provide a measure of inclusion/exclusion consistency between reviewers [48]. The initial kappa score was 0.28. Discrepancies were discussed and resolved after each batch, until the kappa score reached 0.81—which exceeds the recommended threshold of 0.60 [44]—at which point the reviewers screened and coded independently. Double screening was conducted on 100 articles (out of a total of 1352 retrieved at full text) to check for consistency in this manner. All data coding were undertaken in Microsoft Excel, using drop-down options to ensure categorical and consistent data entry. Discrepancies in coding decisions were discussed and resolved in a similar manner to screening discrepancies. For a full list of article meta-data and variables that were coded, see Appendix B Table A2.

No critical appraisal of study internal or external validity was undertaken. This is consistent with CEE guidelines as the purpose of the map is to document the evidence base rather than measure

effect sizes through a meta-analysis. However, it is important to note that studies were rejected where elements of our inclusion criteria were missing, so this evidence map has a high degree of filtering in place that ensures that only studies with clearly reported study designs and results were included.

All figures were plotted in R using the *ggplot2* package. The interactive evidence map of the all relevant studies was created using the Thalloo mapping framework [49]. We used these to assess knowledge gaps and clusters in the evidence and observe any trends in study findings. We have been careful not to over-interpret the number of "recorded outcomes" in the evidence map, given that no statistical analyses were undertaken. Our analyses were limited to answering the simple question "is there any evidence of an effect", not what is the size of the effect [50].

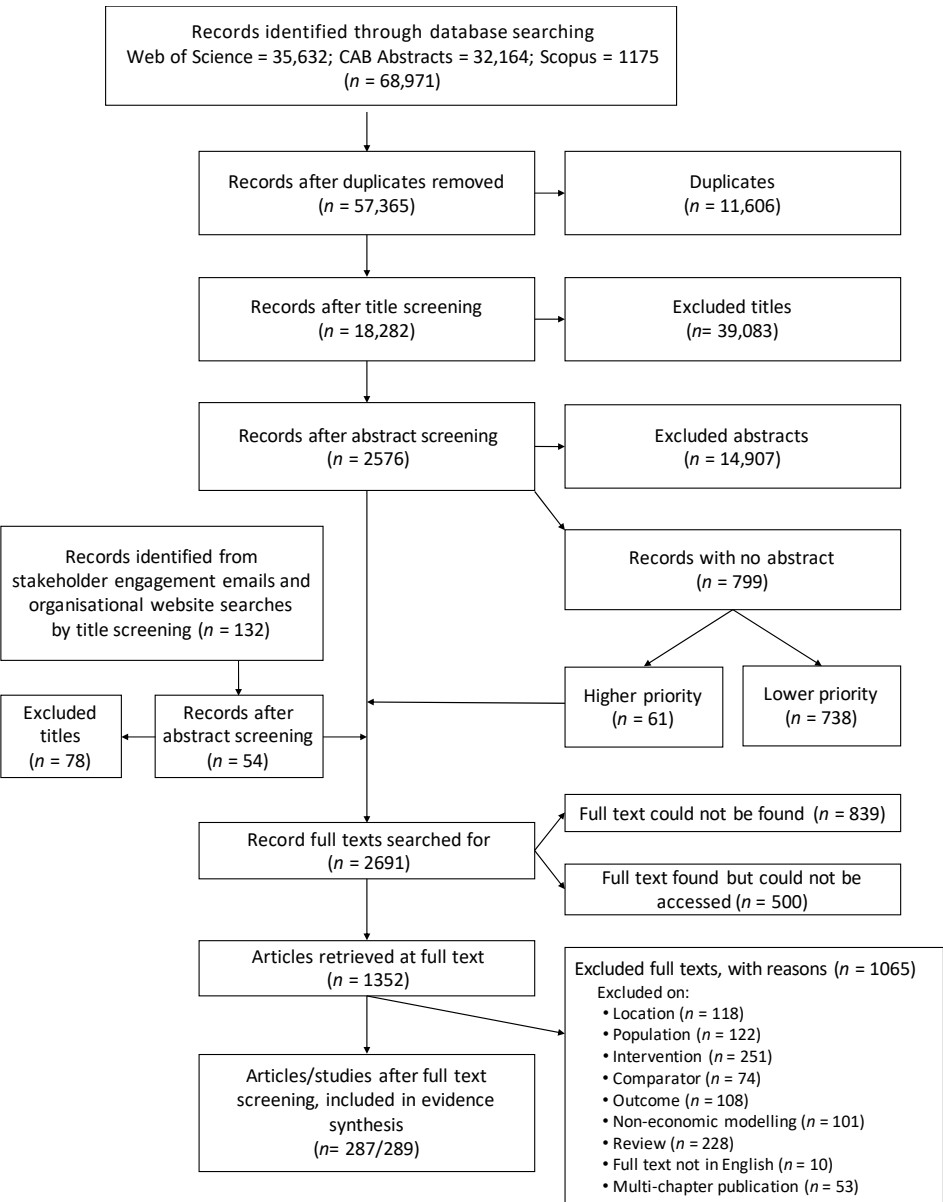

**Figure 1.** The literature searching and screening process, with the number of records included and excluded at each stage. For articles excluded at the full text screening stage, the number excluded for each reason is also given. Two full-text-relevant articles were separated into two studies each, hence the close similarity between the number of articles/studies included in the evidence synthesis. This flow chart follows the template of Haddaway NR [51].

## 3. Results and Discussion

### 3.1. Search Results

From 68,971 records identified through searches of bibliographic databases, and a further 132 identified from organisational websites and stakeholder engagement, 287 articles (289 studies) were screened as relevant at full-text screening for inclusion in this evidence database (Figure 1). Details of the 2692 articles that full texts were searched for are recorded in Supplementary Table S1. The evidence database of included studies is mapped at https://oxlel.github.io/evidencemaps/agroforestry_main/.

Separate maps are available for productivity, GHGEs/carbon and economics showing recorded outcomes (links in relevant results sections below). These interactive maps and accompanying databases can be filtered and queried by publication year, country (location), livestock type (population), agroforestry system (intervention), comparator and productivity, environmental or economic outcomes to provide the searcher with the desired literature (see the help file in the online map descriptor for further details).

### 3.2. Study Spatial and Temporal Distribution

We identified relevant studies in 22 countries across temperate regions of Europe, North and South America and Australasia (Table 4, Figure 2). The total number of studies considering cattle and sheep are approximately equal (90 and 89 respectively), although this proportion fluctuates between countries (Table 4). The publication year of relevant studies indicates an exponential increase in the number of publications, in line with a broader trend in the scientific literature [52]. The earliest relevant record was published in 1942, but 90% of relevant studies were published in or after 1990.

**Table 4.** Number of studies included in evidence map separated by country of study (Location) and livestock type (Population), ordered in descending frequency.

| Country of Study | Livestock Type | | | | | Total |
| | Cattle | Sheep | Mixture of Sheep and Cattle | Unclear from Study | Pasture without Livestock [†] | |
|---|---|---|---|---|---|---|
| USA | 41 | 14 | 1 | 5 | 10 | **71** |
| New Zealand | 4 | 10 | 23 | 7 | 8 | **52** |
| Australia | 4 | 23 | 7 | 8 | 4 | **46** |
| UK | 6 | 17 | 3 | 5 | - | **31** |
| Spain | 1 | 7 | 3 | 3 | 7 | **21** |
| Canada | 8 | 1 | - | 2 | - | **11** |
| Portugal | 2 | 5 | - | 2 | 1 | **10** |
| Chile | 6 | 3 | - | - | - | **9** |
| Argentina | 5 | - | - | 2 | 1 | **8** |
| Italy | 2 | 2 | 2 | 3 | - | **9** |
| France | 1 | 3 | - | - | - | **4** |
| Greece | - | 2 | - | - | 1 | **3** |
| Belgium | 3 | - | - | - | - | **3** |
| Netherlands | 1 | - | - | 1 | - | **2** |
| Austria | 2 | - | - | - | - | **2** |
| Finland | 1 | 1 | - | - | - | **2** |
| Switzerland | 1 | - | - | - | - | **1** |
| Hungary | - | - | - | - | 1 | **1** |
| Poland | 1 | - | - | - | - | **1** |
| Romania | - | 1 | - | - | - | **1** |
| Sweden | 1 | - | - | - | - | **1** |
| **Total** | **90** | **89** | **39** | **38** | **33** | **289** |

[†] Studies of pasture without livestock were searched for and included (Appendices A.1 and B.1.1) in order to find studies that quantify pasture production under agroforestry without referring specifically to livestock types.

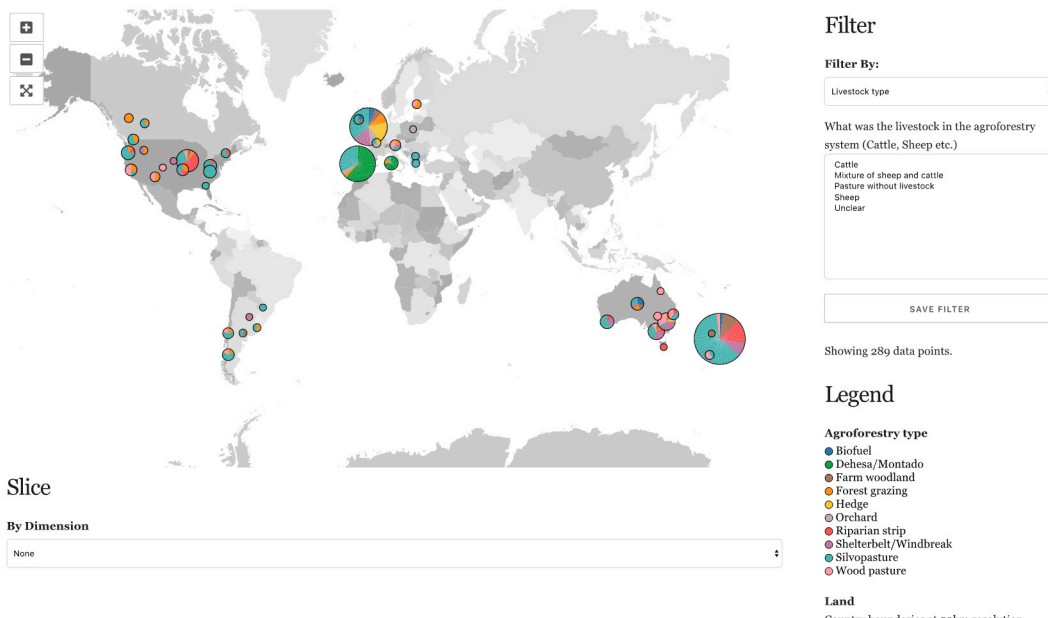

**Figure 2.** Map of 289 relevant studies included in evidence synthesis. Position of pie charts reflects study locations (degrees decimal coordinates), size of pie charts is proportional to the number of studies in that region (or the site when zoomed in online), and the colour of the chart segments shows the number of studies of each agroforestry type (see legend). An interactive version of this evidence map with the accompanying database is available online.

The temperate agroforestry evidence base is predominantly from a small number of countries; 76% of studies in our map are from the USA, New Zealand, Australia, the UK or Spain (Table 4). Although further research in relatively underrepresented temperate countries (e.g., much of Europe) would be useful in providing locally relevant and context-specific evidence, agroforestry trials are expensive to establish and take a long time to mature. Therefore, it could be more effective to investigate how relevant and applicable evidence from well-studied countries such as the USA or New Zealand is to contexts and countries with much less existing research. This could complement investment in new trials by informing dissemination programmes in the short to medium term.

### 3.3. Outcomes-Productivity

We identified 155 studies that recorded productivity measures in a livestock agroforestry system. Measures of "productivity" in these studies included pasture growth rates or herbage yield (i.e., tree understory or pasture production) and livestock growth rates, milk yield, mortality and heat- or cold-stress. Twenty-four studies included both a pasture-related measure and a livestock-related measure, and so were included as separate records, resulting in 179 records in the evidence database, mapped with recorded outcomes at https://oxlel.github.io/evidencemaps/agroforestry_productivity/.

Most records (58%) compared the impacts of agroforestry with pasture without trees. Of 179 records, 51% measured pasture production, 24% measured livestock growth, 17% measured total pasture plus tree production, with the other livestock measures (mortality, heat and cold stress and milk yield) comprising the remainder. Pasture production, livestock growth and total pasture plus tree production were most commonly measured in silvopasture (60%) and forest grazing (13%) systems, whereas most (64%) measures of livestock mortality, heat and cold stress and milk yield were from shelterbelt or windbreak systems.

Although there is a substantial evidence base on the impact of agroforestry systems on pasture production and livestock growth rates, other measures of livestock productivity have received much less attention. This is problematic because one of the key incentives for temperate livestock farmers

to uptake agroforestry is considered to be the benefits of shade or shelter on livestock heat and cold stress and mortality [53]. While all bar one study of livestock mortality, milk yield or heat and cold stress identified by this map found that agroforestry had a positive effect, this is from a total of only 14 studies, which includes grey literature case studies as well as empirical peer-review scientific literature. This finding must therefore be treated with caution until further studies emerge. This number of studies is also too small to confidently identify the factors (e.g., climatic conditions, tree species, tree planting density, etc.) that underpin the apparent variation in magnitude and direction of effects that is seen between studies.

In terms of negative impacts, 53% of studies on pasture production found that agroforestry either had an outright negative effect on production or that it showed an incremental decrease with increased tree density (stems per ha), cover (% canopy) or proximity to pasture measured. This finding is in line with the conclusions of previous reviews on this topic [39,40]. However, only 20% of livestock growth studies found an outright negative effect. This disparity suggests that pasture production is possibly not the only factor influencing livestock growth, and perhaps other factors like pasture quality (e.g., crude protein content [54]) and the effects of tree shade on reducing livestock heat and cold stress could be important. Although some studies also measured pasture quality, we did not include this in our evidence mapping. Further empirical work is needed to determine the factors that drive differences in livestock growth between agroforestry systems and pasture without trees, in order to establish the conditions under which other benefits of tree presence compensate for reduced pasture production, resulting in an overall positive effect on livestock growth rates.

### 3.4. Outcomes-Environmental Indicators

### 3.4.1. Greenhouse Gas Emissions and Carbon Stocks

We identified 77 studies that quantified the impact of agroforestry with livestock on GHGEs or carbon stocks. Three of these studies included measures of two different carbon stocks in the agroforestry system, so were included as separate records, resulting in 80 records in the evidence database. These are mapped with recorded outcomes at https://oxlel.github.io/evidencemaps/agroforestry_carbon/.

Approximately 75% of these records quantified impacts on soil carbon (soil organic carbon, soil organic matter or total soil carbon), of which 42% were in silvopasture systems and 27% in *dehesa* or *montado* systems. When compared to pasture without trees, soil carbon was often found to be higher in agroforestry systems (see map link directly above). This accords with the mechanisms reviewed by Lorenz and Lal [41], which include the potential of the tree component of agroforestry systems to increase inputs of belowground carbon through root litter and exudates, and improve productivity and therefore soil carbon sequestration through nutrient and water uplift from deeper soils inaccessible to herbage. There is also some evidence that implementing silvopastoral systems can result in higher total carbon stocks (soil plus tree biomass) than if the equivalent number of trees was planted as woodland separate to pasture [55–57].

Only six studies explicitly examined the potential of agroforestry to offset or mitigate the GHGEs associated with ruminant livestock [15,16,58–61]. All demonstrated that agroforestry systems have the potential to mitigate at least some of the emissions associated with the livestock component through carbon sequestration in the tree component. However, meta-analyses and modelling of carbon sequestration of different agroforestry systems would need to be coupled with context-specific lifecycle assessments of livestock production under each of these systems in order to demonstrate an empirical baseline in which agroforestry offers an effective and/or complete livestock GHGE mitigation strategy.

Although there are several viable technical options to mitigate ruminant-associated emissions [62–64], using agroforestry practices to offset remaining emissions via compensatory carbon sequestration in woody biomass and soil is not without challenges or controversy. For example, the carbon sequestration potential of agroforestry systems is likely to saturate over time [41], through a combination of the soil carbon pool reaching a new equilibrium and tree growth slowing or stopping

upon maturity. To achieve continuous carbon sequestration, livestock farmers would need to manage trees in a harvesting and restocking cycle, with a guarantee that the end-use of the timber harvested was not going to result in the release of the stored carbon (e.g., by use in construction rather than bioenergy without carbon capture and storage). Furthermore, methane from eructation (belching) and manure forms the majority of emissions associated with temperate ruminant production [5]. Because this is a short-lived gas that reaches an equilibrium concentration in the atmosphere under stable emissions, this has been demonstrated to be equivalent in terms of warming to net zero emissions of carbon dioxide [65]. Although this has led some to question whether stable methane emissions need to be offset at all, others contend that methane offers a "particularly attractive target gas for short-term climate change mitigation" [66].

In this systematic evidence map we focused on carbon sequestration, as the most direct mechanism by which agroforestry systems could mitigate the climate change impacts of GHGEs associated with ruminant production, and did not consider non-$CO_2$ greenhouse gases. However, nitrous oxide emissions are a key environmental impact associated with agricultural generally (predominantly via synthetic fertiliser applications) [67], and ruminant livestock specifically (via manure production and management), which receives substantial policy attention. It is clear that inclusion of trees in agricultural systems via agroforestry can affect nitrous oxide emissions [68–71] and although a recent meta-analysis found no clear overall direction of effect [72], this should be included in future evidence syntheses.

### 3.4.2. Water Quantity

Twenty-two studies quantified the effect of an agroforestry intervention on water quantity (by measuring water runoff, infiltration or hydraulic conductivity) across six of the 22 countries covered by this systematic map. Of these, approximately half (55%) found agroforestry to have a significant positive effect including lower runoff and higher soil infiltration capacity.

Although planting trees is widely held to be an effective for flood hazard reduction, this is increasingly recognised as a simplistic view [73], with a recent meta-analysis finding tree planting to have a modest but highly variable impact on channel discharge [42]. Moving from the catchment to the field scale, research on the potential of trees planted in agroforestry systems to mitigate any increase in surface runoff caused by livestock trampling indicates that the presence/absence of livestock is likely to be as important, if not more so, than the presence/absence of trees [21,74,75]. This suggests that linear field-edge agroforestry systems such as riparian strips and shelterbelts where livestock are excluded hold more potential for mitigating any livestock-induced runoff from pasture than systems with livestock grazing underneath trees such as silvopasture.

### 3.4.3. Water Quality

Twenty-seven studies quantified the effect of an agroforestry intervention on water quality, through measuring water sediment or nutrient concentrations. Riparian strips were by far the most common agroforestry system in water quality studies.

Although 16 of the 27 water quality studies found a significant positive influence of the agroforestry intervention, nine of the studies in riparian strips found no or mixed effects on water quality. This variation in study findings is likely to be influenced at least in part by factors such as buffer understory and width [76,77]. While tree roots can play an important role in stabilising watercourse banks [78], grass buffers have been found to be more effective in capturing sediments and diffuse pollutants in field runoff, leading to zoned buffers comprising a woody component and a separate uphill grass component being proposed as a potential best practice to compensate for the shading effects of the tree component on understory composition [76,79]. However, the need for wide buffers may deter farmer uptake [80] and could limit any shelter or shade benefits to livestock from trees in an agroforestry context. Future research on woody riparian strips should investigate differing tree and shrub planting densities to

establish integrated designs that maintain a sufficient grass understory to maximise nutrient and sediment removal at more appealing buffer widths.

### 3.4.4. Soil Erosion

Twenty-eight studies quantified the effect of an agroforestry intervention on soil erosion across five countries. Of these, 71% found a significant positive effect of the intervention. This finding is expected given that tree planting is widely used as an appropriate strategy to stabilise slopes and reduce soil erosion globally. In particular, widespread planting has occurred on Australian and New Zealand livestock farms in recent decades in attempts to reverse the negative consequences of extensive vegetation clearing following European settlement [9]. Only two studies found a significant negative effect of agroforestry on soil erosion in our map. These were studies of forest grazing with ungrazed forest as a control, so it is unsurprising that the addition of livestock trampling to forest resulted in an increase in erosion, and does not negate the evidence supporting a positive influence of trees planted on pasture.

### 3.4.5. Air Quality

We did not identify any studies that collected primary field data on the impact of an agroforestry intervention on air quality from a ruminant farming system. Although two air quality studies were included on the search test list (Appendix A.4), these were excluded at the full-text screening stage due to not meeting the inclusion criteria. Bealey et al. [81] was a modelling study with no primary data collection and Lin et al. [82] looked at odour dispersion rather than air pollutants such as ammonia. Empirical data from field trials is required if agroforestry systems are to be recommended as a means to mitigate ruminant ammonia emissions.

### 3.5. Outcomes—Economics

Sixty-four studies quantified or modelled the impact of integrated livestock and trees (or other woody perennials) on farm or forestry enterprise economics https://oxlel.github.io/evidencemaps/agroforestry_economics/.

Of these, the majority (73%) came from the four most represented countries (New Zealand, Australia, UK, USA) (Figure 3). Three-quarters of studies found a positive impact of integration on enterprise economics, compared to separate livestock and trees. However, this evidence base comprises a diversity of study types, including case studies of return on investment from establishing an agroforestry system, mathematical modelling of optimal densities of trees and livestock under different commodity prices, and cash flow analyses. Although most of these studies focused on income streams from livestock, timber or non-timber tree products, as payments for ecosystem services become increasingly available—particularly in the form of carbon credits for $CO_2$ sequestration—this is likely to further increase the economic appeal of agroforestry compared to conventional agriculture [83,84].

### 3.6. Trade-Offs and Synergies

Our extraction of the evidence-base from these 289 studies enables the exploration of trade-offs and synergies between multiple environmental and productivity indicators, allowing previous work in this area such as Beckert et al. [85] to be built upon. Here we examine the relationship between agroforestry systems and a) soil carbon stocks, b) pasture production, c) livestock growth and d) heat and cold stress, milk yield and mortality, for studies with a "pasture without trees" comparator (Figure 4). For ease of visualisation, agroforestry systems were grouped according to similarity into the categories of: (i) silvopasture, (ii) *dehesa*, *montado*, wood pasture or forest grazing (i.e., more semi-natural systems) and (iii) shelterbelt, windbreak, riparian strip, hedge (i.e., linear field-edge systems). Relevant records from biofuel and farm woodland systems (two for soil carbon, three for pasture production, three for livestock growth) are not included in Figure 4.

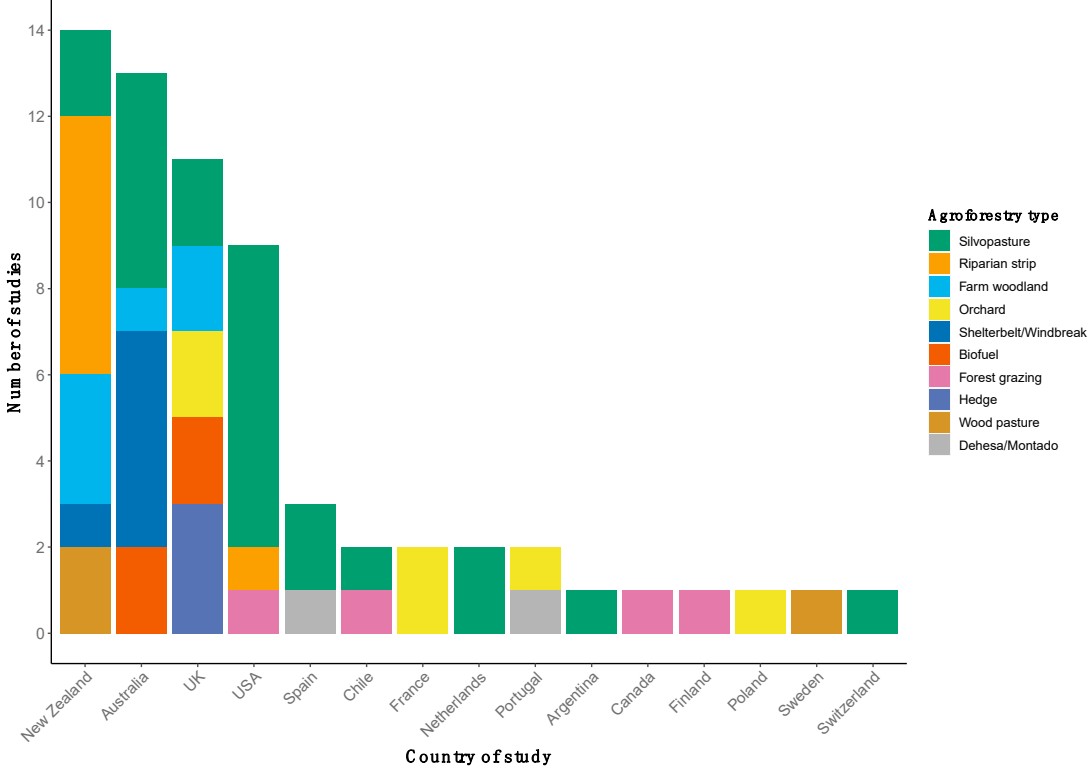

**Figure 3.** Studies that quantified or modelled the impact of agroforestry on enterprise economics (*n* = 64) compared to one or multiple non-integrated systems, separated by agroforestry type (colours within stacked bars) and country of study.

Considering semi-natural systems (*dehesa*, *montado*, wood pasture and forest grazing) compared to pasture without trees, Figure 4a suggests that these systems typically have a positive effect on soil carbon stocks. There is an apparent trade-off with pasture production (Figure 4b); no studies in our map found improved pasture growth under these systems compared to open pasture. However, livestock growth rates are typically improved in these agroforestry systems compared to open pasture (Figure 4c). This may indicate that there are other factors important for livestock growth, such as reduced temperature stress or improved pasture quality, that compensate for reduced pasture production. However, we found no studies of livestock heat or cold stress in these systems (missing panel in Figure 4d), and we did not search for measures of pasture quality in this systematic map. Nevertheless, this suggests that a valuable future research contribution would be to model livestock growth in these systems compared to open pasture accounting for pasture production and quality and measures of livestock temperature stress. This could then be used to see if these productivity relationships are applicable to other temperate regions outside of the Mediterranean-type climate that many of these studies are from.

Similarly, linear systems (shelterbelts and windbreaks) appear to have a positive influence on livestock growth, heat stress, cold stress, milk yield and weather-related mortality (Figure 4c,d). However, there is much less evidence on the influence of these systems on pasture productivity (Figure 4b), a knowledge gap identified over 20 years ago [32], or soil carbon stocks (Figure 4a). Such field-edge agroforestry systems are also potentially suitable to improve water quantity regulation and water quality (see Sections 3.4.2 and 3.4.3). Therefore, further research into a complete basket of productivity and environmental measures would be valuable for these systems, to quantify trade-offs and synergies in delivery of these.

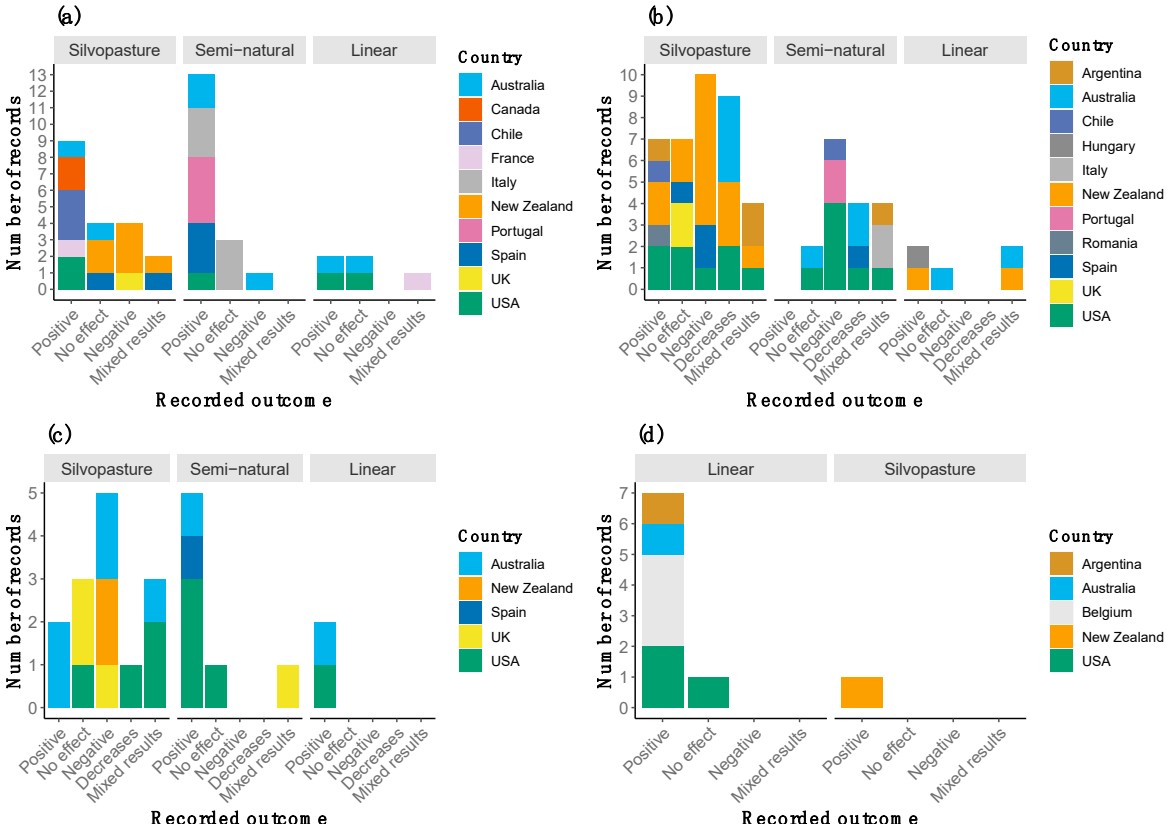

**Figure 4.** Studies with a "pasture without tree" comparator that recorded (**a**) soil organic carbon, soil organic matter or total soil carbon (*n* = 41), (**b**) pasture production (*n* = 51), (**c**) livestock growth (*n* = 23) and (**d**) livestock heat stress, cold stress, mortality or milk yield (*n* = 9). Colours of stacked bars indicate country of study. Panel headings indicate the agroforestry systems included in each facet: "Silvopasture" refers to trees in wide-spaced rows or clumps with herbage growing underneath; "Semi-natural" refers to *dehesa*, *montado*, wood pasture or forest grazing systems; and "Linear" refers to shelterbelts, windbreaks, riparian strips and hedgerows. The small number of studies in farm woodland or biofuel systems are not included. In (**b**,**c**), the recorded outcome "Decreases" was used for studies which found decreased pasture production or livestock growth, respectively, across a gradient of increased tree density, cover or proximity, whereas "Negative" refers to lower production or growth under agroforestry compared to pasture without trees.

*3.7. Limitations*

Although our systematic map covered most major temperate regions of the world, we only included studies written in English. Much of the peer-reviewed scientific literature is now published in English, but we anticipate that this had some impact on our grey literature search. We are aware that additional search terms could have yielded more relevant studies, but they would also have included many more studies of no relevance, thereby disproportionately increasing the screening effort needed to identify relevant studies; the test set of papers gave us confidence that our search strategy balanced precision with accuracy. We based our search of temperate regions based on the countries practicing temperate agroforestry included in Gordon, Newman and Coleman [13]. However, it is apparent that "country" provides an incomplete level of climatic resolution, and we therefore recommend that further meta-analyses using this database additionally codes studies using a climate classification such as the Köppen-Geiger [86]. Furthermore, we are aware that there are areas in India and China that are temperate with relevant agroforestry literature [13], and would suggest that studies from these areas are incorporated, although we recognise that this would require substantial additional effort to identify such studies from the very large body of non-temperate literature from these two countries.

We excluded all papers that considered the impacts of afforesting or reforesting pasture as this constitutes replacing livestock with trees and not integrating them, therefore this is not agroforestry. However, some outcomes (e.g., changes in soil carbon stocks) from these studies are likely to be similar and applicable to some agroforestry systems such as small farm woodlands and shelterbelts/windbreaks. We also excluded all studies that considered shrub or tree encroachment onto rangeland because this was deemed to be not an intentional integration of woody perennials with livestock. However, the outcomes (e.g., pasture productivity, carbon stocks) under these systems are arguably applicable to some temperate agroforestry systems.

## 4. Conclusions

We identified, through a systematic mapping process, a substantial evidence base (289 studies) on the productivity, environmental and economic impacts of integrating agroforestry into temperate sheep and cattle farming systems, creating an interactive resource with applications across temperate regions. It is clear that implementation of agroforestry has the potential to sequester carbon, reduce soil erosion, and, with appropriate management, improve water quantity and quality regulation. However, the impact of agroforestry implementation on pasture productivity and livestock growth are variable, and livestock productivity measures such as heat and cold stress and weather-related mortality have been little-studied. Although we suggest some possible trade-offs and synergies between agroforestry system types and the delivery of environmental and productivity benefits, meta-analyses are required to validate these suggested trends for already well-studied outcomes, complemented by further field research where data is inadequate. Although widespread economic benefits of agroforestry adoption are reported in the literature, studies are generally highly context-specific. This systematic map can be applied across temperate regions by researchers, policymakers and practitioners as a resource to inform promotion and implementation of agroforestry practices that increase the sustainability of ruminant livestock production.

**Supplementary Materials:** The following are available online at http://www.mdpi.com/1999-4907/11/12/1321/s1, Table S1: Agroforestry systematic map full text screening and data coding.

**Author Contributions:** Conceptualisation, M.W.J., K.J.W. and G.P.; methodology, M.W.J. and G.P.; software, M.W.J. and L.P.; validation, M.W.J., W.J.H., L.P. and G.P.; formal analysis, M.W.J.; investigation, M.W.J., W.J.H. and L.P.; resources, M.W.J., W.J.H. and L.P.; data curation, M.W.J., W.J.H. and L.P.; writing—original draft preparation, M.W.J.; writing—review and editing, M.W.J., K.J.W. and G.P.; visualization, M.W.J. and L.P.; supervision, K.J.W. and G.P.; project administration, M.W.J.; funding acquisition, M.W.J. All authors have read and agreed to the published version of the manuscript.

**Funding:** This research was funded by Biotechnology and Biological Sciences Research Council (BBSRC), grant number BB/M011224/1.

**Acknowledgments:** We would like to thank members of the UK Farm Woodland Forum and European Agroforestry Federation for valuable feedback on the mapping proposal via their online mail lists. We would also like to thank the reviewers for their helpful comments.

**Conflicts of Interest:** The authors declare no conflict of interest. The funders had no role in the design of the study; in the collection, analyses, or interpretation of data; in the writing of the manuscript, or in the decision to publish the results.

## Appendix A. Search Strategy

*Appendix A.1. Search String*

sheep OR ewe* OR lamb OR lambs OR lambing OR "Ovis aries" OR ovine OR "Bos taurus" beef OR dairy OR cattle OR cow OR bull OR steer OR heifer OR cows OR bulls OR steers OR heifers OR calf OR calves OR calving OR bovine OR grazed OR grazing OR graze OR ruminant OR ruminants OR livestock OR pasture OR pastures OR pastoral

AND

tree OR trees OR shrub OR shrubs OR shrubby OR wood OR woodland OR woods OR woodlot* OR forest OR forests OR forestry OR silvopast* OR sylvopast* OR silvipast* OR sylvipast* OR agriforest* OR

agroforest* OR silvo-past* OR sylvo-past* OR silvi-past* OR sylvi-past* OR agri-forest* OR agro-forest* OR agrosilvopast* OR agrisilvopast* OR agrosylvopast* OR agrisylvopast* OR agrosilvipast* OR agrisilvipast* OR agrosylvipast* OR agrisylvipast* OR "wood past*" OR dehesa OR dehesas OR montado OR montados OR bocage OR bocages OR shelterbelt OR shelterbelts OR "wind break*" OR windbreak* OR "riparian buffer*" OR "riparian strip*" OR "buffer strip" OR hedge OR hedges OR hedging OR orchard OR orchards OR "multipurpose tree*" OR intercrop* OR "alley crop*" OR "row system" OR "row systems" OR "clump system" OR "clump systems" OR "linear feature" OR "linear features" OR biofuel OR biofuels OR bioenergy OR coppic* OR "short rotation woody crop*" OR "short-rotation forest*" OR fuelwood OR "fuel wood"

　　AND

"environment* benefit*" OR "environment* impact*" OR externality OR externalities OR "greenhouse gas*" OR GHG OR GHGE OR GHGs OR GHGEs OR offset* OR mitigat* OR sequest* OR emission* OR abate OR abates OR abating OR abatement OR carbon OR "air pollut*" OR methane OR "nitr* oxide" OR "nitr* oxides" OR "nitr* dioxide" OR "nitr* dioxides" OR "climate chang*" OR "global warm*" OR flood* OR "water flow regulat*" OR hydrolog* OR infiltrat* OR "water quality" OR "water puri*" OR "dissolved solids" OR "suspended solids" OR sediment* OR nutrient* OR "run off" OR runoff OR fertilis* OR fertiliz* OR manur* OR pollut* OR nitrogen OR nitrate OR phosphorous OR phosphorus OR phosphate OR "air qualit*" OR ammonia OR ammonium OR odo*r OR erosion OR erode OR eroding OR eroded OR erodes OR "soil loss*" OR "soil degrad*" OR productivity OR production OR "animal health" OR "animal welfare" OR "shade" OR shelter OR "heat stress" OR "cold stress" OR "tree-animal synerg*" OR "cattle cool*" OR "sheep cool*" OR "animal perform*" OR liveweight OR bioclimat* OR "growth rate" OR pasture OR pastures OR "wind protect*" OR microclimate* OR yield OR "carrying capacity" OR financ* OR economic* OR "bio-economic model*" OR "cost-benefit analysis" OR diversif*

　　AND

Temperate OR "UK" OR "United Kingdom" OR England OR Scotland OR Wales OR "Northern Ireland" OR English OR Welsh OR Scottish OR Irish OR Europ* OR Germany OR France OR Italy OR Spain OR Ukraine OR Poland OR Romania OR Netherlands OR Holland OR Belgium OR Czech* OR Greece OR Portugal OR Sweden OR Hungary OR Belarus OR Austria OR Serbia OR Switzerland OR Bulgaria OR Denmark OR Finland OR Slovakia OR Norway OR Ireland OR Croatia OR Moldova OR Bosnia OR Albania OR Macedonia OR Slovenia OR Latvia OR Estonia OR Montenegro OR Luxembourg OR Malta OR Iceland OR Andorra OR Monaco OR Liechtenstein OR "San Marino" OR "Channel Islands" OR "Isle of Man" OR Gibraltar OR "Faroe Islands" OR Yugoslavia OR German OR French OR Italian OR Spanish OR Ukrainian OR Polish OR Romanian OR Dutch OR Belgian OR Czech* OR Greek OR Portuguese OR Swedish OR Hungarian OR Belarusian OR Austrian OR Serbian OR Swiss OR Bulgarian OR Danish OR Finnish OR Slovakian OR Norwegian OR Croatian OR Moldavian OR Bosnian OR Albanian OR Macedonian OR Slovenian OR Latvian OR Estonian OR Montenegrin OR Luxembourgish OR Maltese OR Icelandic OR Andorran OR Monegasque OR Sammarinese OR Manx OR Yugoslav OR Canada OR "USA" OR "North America" OR "United States of America" OR "US" OR Canadian OR American OR Chile OR Patagonia OR Argentina OR Chilean OR Patagonian OR Argentinean OR Australi* OR "New Zealand"

Appendix A.1.1. Further Details

An identical search string was used to search Web of Science, Scopus and CAB Abstracts on 2 October 2019, with the exceptions of the removal of the "US" term in CAB Abstracts and the addition of "TITLE-ABS-KEY" syntax required by Scopus. Web of Science was searched across "All Databases" which included:

- Web of Science Core Collection (1900–present)

  ○　　Science Citation Index Expanded (1900–present)

- ○ Social Sciences Citation Index (1900–present)
- ○ Arts & Humanities Citation Index (1975–present)
- ○ Conference Proceedings Citation Index-Science (1990–present)
- ○ Conference Proceedings Citation Index-Social Science & Humanities (1990–present)
- ○ Book Citation Index–Science (2005–present)
- ○ Book Citation Index–Social Sciences & Humanities (2005–present)
- ○ Emerging Sources Citation Index (2015–present)
- ○ Current Chemical Reactions (1986–present) *(Includes Institut National de la Propriete Industrielle structure data back to 1840)*
- ○ Index Chemicus (1993–present)
- BIOSIS Citation Index (1969–present)
- Current Contents Connect (1998–present)
- Data Citation Index (1993–present)
- Derwent Innovations Index (1993–present)
- KCI-Korean Journal Database (1980–present)
- MEDLINE® (1950–present)
- Russian Science Citation Index (2005–present)
- SciELO Citation Index (2002–present)
- Zoological Record (1993–present)

*Appendix A.2. Stakeholder Engagement*

Emails were sent on 2 October 2019 to the following temperate agroforestry organisations, using contacts available in Gordon, Newman and Coleman [13]:

- Farm Woodland Forum-UK-JISC maillist
- European Agroforestry Federation (EURAF)-euraf@agroforestry.eu
- Australian Agroforestry Foundation-admin@agroforestry.org.au
- Association for Temperature Agroforestry-North America-online form submission
- Poplar and Willow Research Trust-New Zealand-ian.mcivor@plantandfood.co.nz

The following email was circulated:

Dear concerned,

I am a PhD student at the University of Oxford, United Kingdom, researching the potential for agroforestry to increase the sustainability of sheep and cattle (beef and dairy) production. I am contacting your organisation for feedback on a systematic literature review proposal.

I am in the process of systematically mapping the evidence base to answer the question: "What are the impacts of temperate silvopastoral systems on sheep and cattle productivity, environmental impacts* and farm economic viability?"

* specifically focusing on greenhouse gas emissions, reduced water and air quality, increased flood hazard and enhanced soil erosion attributed to sheep and cattle farming.

I am interested in how agroforestry could mitigate some/all of these. I am following the Collaboration for Environmental Evidence guidelines, and so am contacting relevant stakeholders to:

(1) receive feedback on the applicability and necessity of this work;

(2) determine how best to improve and hone the review question to ensure the findings are most relevant and useful to practitioners; and

(3) request submissions of any studies that you are aware of that are particularly relevant to this question, particularly grey literature that might not be findable using online bibliographic database searches.

I would be very grateful if you could circulate this to your members, with a request for feedback on the review question and/or links to relevant studies, emailed to mattthew.jordon@zoo.ox.ac.uk.

Very best wishes,

Matt Jordon

Replies to this email (exclusively from the UK Farm Woodland Forum and EURAF) provided 14 articles that we could access but were not already present in our search results. These were incorporated into the grey literature screening process.

*Appendix A.3. Organisational Websites Searched for Grey Literature*

Organisational websites were searched over the week beginning 16 March 2020. The websites searched, and number of relevant articles downloaded from each after title screening, is given in Table A1. In total, combined with the articles received through the stakeholder engagement emails (Appendix A.2), this resulted in 132 articles taken to grey literature abstract screening (Main Text Figure 1). Relevant grey literature records that were already present in the "records after abstract screening" (from the bibliographic databases searched) were not counted or included again.

**Table A1.** Organisational websites searched for grey literature.

| Country/Region | Organisation | Website Address | Number of Articles Retrieved |
|---|---|---|---|
| UK | Woodland Trust | https://www.woodlandtrust.org.uk/ | 15 |
| | Forest Research | https://www.forestresearch.gov.uk/ | 15 |
| | Organic Research Centre | http://www.organicresearchcentre.com/ | 4 |
| | ClimateXChange | https://www.climatexchange.org.uk/ | 3 |
| | Farm Woodland Forum | https://www.agroforestry.ac.uk/ | 0 |
| | Game and Wildlife Conservation Trust | https://www.gwct.org.uk/ | 0 |
| | Agricology [†] | https://www.agricology.co.uk/ | 0 |
| | Soil Association [†] | https://www.soilassociation.org/ | 0 |
| Europe | European Agroforestry Federation (EURAF) | https://euraf.isa.utl.pt/welcome | 11 |
| | Agroforestry Innovation Networks (AFINET) | http://www.eurafagroforestry.eu/afinet | 35 |
| | AGroFORestry that Will Advance Rural Development (AGFORWARD) | https://www.agforward.eu/index.php/en/ | 12 |
| | European Forestry Institute (EFI) | https://www.efi.int/ | 0 |
| New Zealand | The New Zealand Poplar and Willow Research Trust | https://www.poplarandwillow.org.nz/ | 23 |
| Australia | Australian Agroforestry Foundation | http://agroforestry.org.au/ | 0 |
| USA | Association for Temperate Agroforestry | https://www.aftaweb.org/ | 0 |

[†] Agricology and the Soil Association contained relevant articles, but these had already been found through the Woodland Trust website.

*Appendix A.4. Test List Used to Estimate Comprehensiveness of Search*

Productivity–[32,39,40,87–91]
Environmental impacts

- GHGEs–[15,16,41,56]
- Flood hazard–[20,21,74]
- Water quality–[18,92]
- Air quality–[81,82]
- Soil erosion–[9,23]

Economics-[93,94]

*Notes on Final Inclusion of Test List Articles*

All of the above articles were found using the search terms given in Appendix A.1. Bealey et al. (2014), Benavides et al. (2009), Bird (1998), Lorenz and Lal (2014) and Mead (1995) were excluded at the full-text screening stage under "Review" because they did not contain any primary data. Hawke et al. (1994) and Lin et al. (2006) were excluded at the full-text stage under "Outcome" and McIvor et al. (2008) were excluded at the abstract screening stage, due to not meeting the inclusion criteria (Appendix B.2). All other studies were screened as relevant at the full-text stage and coded accordingly.

## Appendix B. Article Screening

MWJ conducted all title and abstract screening. WJH and LP searched for all article full texts, apart from grey literature, which was searched for and screened by MWJ. MWJ and WJH shared full text screening and data coding. Articles that met all the PICOL criteria but did not contain any primary data (i.e., literature reviews, meta-analyses, modelling studies) or were stakeholder engagement or land manager survey studies were excluded with the reason given as "Review". Multi-chapter publications (books, conference proceedings, research reports) were coded as such and no data extracted due to time limitations. These citations are recorded in Supplementary Table S1 and further data extraction would be possible in future synthesis work.

*Appendix B.1. Inclusion/Exclusion Criteria*

The criteria used to decide whether articles should be included or excluded at title, abstract and full text screening are given below, ordered by PICOL elements (Population, Intervention, Comparator, Outcome, Location; Main Text Table 2).

Appendix B.1.1. Population

Included

- Sheep, beef cattle, dairy cattle
- Pasture (under trees/shrubs)

Excluded

- Ruminant livestock that are not sheep or cattle (e.g., goats)
- Non-ruminant livestock (e.g., pigs, poultry)
- Studies that solely consider non-forage crops (e.g., silvoarable systems)

Appendix B.1.2. Intervention

Included

- Silvopasture, shelterbelts/windbreaks, riparian strips, hedges, *dehesa/montado*, wood pasture, forest grazing, orchards, woody biofuel and farm woodlands implemented with pasture for livestock grazing as an understory or in the surrounding agricultural matrix

Excluded

- Articles that compare non-grazed forestry/afforestation with pasture; not agroforestry because not integrated
- Articles that consider woody plant/shrub encroachment onto rangeland or wholly natural savannah systems; not agroforestry because the integration of trees and livestock is not intentional
- Riparian buffer strips that did not contain any woody perennials (e.g. grass buffers)
- Non-woody shelterbelts/windbreaks (e.g. use of tall grasses)

- Any artificial manipulations without real trees/shrubs, e.g., shade experiments using shade cloth, windbreak experiments using plastic sheets, addition of leaf litter onto pasture by humans rather than natural litter-fall

Appendix B.1.3. Comparator

Included

- Pasture without trees/shrubs
- Forest without grazing

Excluded

- Unmanaged land of any form
- Forest grazing by game or wildlife rather than domestic livestock
- Comparison of two or more treatments that are not the intervention of interest (even if under an agroforestry system), e.g. different fertiliser applications, different understory species

Appendix B.1.4. Outcome—Productivity

Included

- Understory/pasture productivity (e.g. dry matter production, herbage yields etc.)
- Livestock mortality, growth rates, heat/cold stress, milk yield

Excluded

- Studies that just measure pasture quality (e.g. crude protein or micronutrient contents) without measuring impact on livestock growth rates o.e.
- Livestock diseases or illness made more or less likely due to proximity to trees, e.g. abortion caused by eating pine needles, increased tick burden in forest grazing
- Any articles considering consumption of tree fodder or browse by livestock
- Any articles that just consider tree growth or timber yields under different systems (unless economic implications are explicit, in which case coded under economics, below)

Appendix B.1.5. Outcome-Environmental impacts

Included

- Greenhouse gas emissions: measurements of emissions or sequestration or stocks of carbon in soil or above- or below-ground plant biomass
- Flood hazard: volume of water runoff, infiltration rate of water into soil, hydraulic conductivity of soil
- Water quality: water nutrient concentrations, suspended sediments, etc.
- Air quality: measurement of removal of air pollutants by trees
- Soil erosion

Excluded

- Flood hazard: measurements just of bulk density/penetration resistance/other measurement of compaction where infiltration/runoff only inferred
- Water quality: measurements just of stream fauna (e.g. macroinvertebrates) and water quality only inferred
- Air quality: impact of air pollutants (e.g. ammonia uptake or deposition) on tree health
- Other exclusion reasons

    ○    Salinity studies

    ○    Impacts or risk of fire

    ○    Exclusively measuring soil micro- and macrofaunal (bacteria, fungi, mycorrhizae, invertebrates including ants and dung beetles)

Appendix B.1.6. Outcome-Economics

Included

- Overall profitability of

    ○    (i) farm business with/without agroforestry, or

    ○    (ii) forestry business with/without livestock grazing, including measurements of value of:

    ○    timber/non-timber tree products/biofuel,

    ○    value of forage for livestock grazing,

    ○    sporting benefits

    ○    payments for ecosystem services (e.g. sale of carbon credits)

Excluded

- Livestock damage to trees in agroforest or forest
- Suppression of natural regeneration of trees by livestock grazing or value of removal of herbage by livestock to facilitate natural regeneration of trees

Appendix B.1.7. Location

Included

- Temperate European countries (see search string, Appendix A.1.)
- New Zealand and temperate regions of Australia
- Temperate regions of North America (parts of Canada and the USA) and South America (parts of Chile and Argentina)

Excluded

- Tropical, sub-tropical, boreal and subarctic/sub Antarctic regions of countries in search string

    ○    Subarctic/boreal regions of Canada and parts of some northern European countries

    ○    Sub-Antarctic regions of Chile and Argentina

    ○    Tropical and sub-tropical regions of Australia (e.g. Queensland), USA (Texas, Oklahoma, Louisiana, Arkansas, Alabama, Mississippi, Tennessee, Kentucky, North Carolina, South Carolina, Georgia and Florida), Chile and Argentina

- Temperate regions in countries not searched for (e.g. parts of China, India, Mexico, Turkey, Russia)

We searched for literature from countries with significant temperate agroforestry research following Gordon, Newman and Coleman [13], with the exception of China and India. We chose to exclude temperate regions in countries not explicitly searched for to avoid the risk of including only part of the evidence bases from these countries. We did not search for every country with a temperate region because study region or coordinates are often not included in article metadata (keywords, title or abstract) and therefore are difficult to accurately search for using bibliographic databases and citation indexes. The alternative of including all countries with a temperate region and then excluding non-temperate studies was deemed too burdensome for a likely relatively low return of relevant studies.

Appendix B.1.8. Language

Included

- English

Excluded

- All others

Appendix B.1.9. Data Range

Included

- All

Excluded

- n/a

*Appendix B.2. Article Meta-Data and Variables Coded*

Fields of article meta-data and variables coded during full text screening and data extraction are given in Table A2. Screening information and article meta-data were coded for all articles screened at full text, whereas PICOL variables were only coded for relevant full-text articles. For the outcome variables, a coding of "positive" indicates that the agroforestry treatment (i.e., the coded Intervention) resulted in a more favourable outcome than the coded Comparator (for example, improved pasture or livestock productivity, less GHGEs, more carbon sequestered, less water runoff, better water or air quality, reduced soil erosion, better enterprise economics), and vice-versa for a coding of "negative", and so on.

**Table A2.** Fields of article meta-data and variables coded during full text screening and data extraction.

| Section | Field | Options |
|---------|-------|---------|
| Screening information | Full text found? <br> Screener <br> Full text relevant? <br> Exclusion reason | Yes/No/Found but cannot access <br> Review 1/Reviewer 2 <br> Yes/No/Book/Text not in English <br> L/P/I/C/O/Review |
| Article meta-data | Year published <br> Author <br> Title <br> Journal Title (if applicable) <br> ISBN/ISSN <br> DOI <br> Volume <br> Issue <br> Pages | |
| Location | Country <br> State <br> Latitude, Longitude (decimal degrees) | |
| | Population | Sheep/Cattle/Mixture of sheep & cattle/Pasture without livestock/Unclear |
| | Intervention [†] | Silvopasture/Shelterbelt/Windbreak/Riparian strip/Hedge/*dehesa*/*montado*/Wood pasture/Forest grazing/Orchard/Biofuel/Farm woodland/Multiple/Other |
| | Comparator | Pasture without trees/Forest without grazing/Baseline from implementation/No comparator/Other |

**Table A2.** *Cont.*

| Section | Field | Options |
|---|---|---|
| Outcome-productivity | Productivity measure | Understory or pasture production/Livestock mortality/Livestock growth/Livestock heat stress/Livestock cold stress/Milk yield/Total productivity (pasture + trees)/Multiple/Other |
| | Effect on productivity measure | Positive/Negative/No effect/Mixed results (positive + no effect)/Mixed results (negative + no effect)/Mixed results (positive + negative)/Decrease with increased tree density or cover or proximity/Unclear |
| Outcome-greenhouse gas emissions and carbon stocks/sequestration | Carbon/GHGE measure | Soil organic carbon/Soil organic matter/Total soil carbon/Belowground carbon (plant roots)/Aboveground carbon (plant stems)/Total carbon (above-and below-ground + soil)/GHGEs/Livestock offset or mitigate/Multiple/Other |
| | Effect on carbon/GHGE measure | Positive/Negative/No effect/Mixed results (positive + no effect)/Mixed results (negative + no effect)/Mixed results (positive + negative)/Unclear |
| Other environmental outcomes | Effect on flooding/runoff/infiltration | Positive/Negative/No effect/Mixed results (positive + no effect)/Mixed results (negative + no effect)/Mixed results (positive + negative)/Unclear |
| | Effect on water quality | Positive/Negative/No effect/Mixed results (positive + no effect)/Mixed results (negative + no effect)/Mixed results (positive + negative)/Unclear |
| | Effect on air quality | Positive/Negative/No effect/Mixed results (positive + no effect)/Mixed results (negative + no effect)/Mixed results (positive + negative)/Unclear |
| | Effect on soil erosion | Positive/Negative/No effect/Mixed results (positive + no effect)/Mixed results (negative + no effect)/Mixed results (positive + negative)/Unclear |
| Outcome-economics | Effect on enterprise economics | Positive/Negative/No effect/Mixed results (positive + no effect)/Mixed results (negative + no effect)/Mixed results (positive + negative)/Unclear |

† Agroforestry types were typically coded as they were described in the underlying article. Although the term "silvopasture" is commonly used as a catch-all term for livestock agroforestry (i.e., intentionally integrating woody vegetation with livestock farming [47]), in most of the literature, "silvopasture" is used to refer more specifically to regularly spaced trees in rows or sometimes clumps. This more specific usage is retained here and separate terms are used to refer to other types of livestock agroforestry.

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
