# Peer review of "Implications of Temperate Agroforestry on Sheep and Cattle Productivity, Environmental Impacts and Enterprise Economics. A Systematic Evidence Map"

_forests, doi:10.3390/f11121321_

Round 1

Reviewer 1 Report

The authors achieved the goal of establishing a database of studies to provides a resource to support context-specific decision making by land managers, academics and policy makers across temperate regions, whilst identifying future field-based research priorities and enabling further quantitative meta-analysis.

The study identifies significant gaps in terms of results in certain agroforestry sectors, including livestock mortality, milk yield and heat and cold stress. They also identified an interesting disparity in results between pasture production and livestock growth. This may stimulate particular examination of this relationship. Also, the study highlights the questions still pending on how agroforestry can affect nitrous oxide emissions, which tree densities is suitable in woody riparian strips, air quality.

It is understood that the direct benefits to producers are less clearly demonstrated than several beneficial effects on environmental conditions. It was stressed that payments to producers for environmental goods and services, such as carbon fees, encourage producers.

This work seems to me very relevant for the publication

Others comments

L79-80 What are the impacts of temperate agroforestry systems on sheep and cattle productivity,  environmental impacts conditions, and farm economic viability?”.

L106 ROSE …..first apparition add….RepOrting standards for Systematic Evidence Syntheses

Suggestion for a better link with Fig 4…

L358 Considering semi-natural systems (dehesa, montado, wood pasture and forest grazing systems) compared to pasture without trees, Figure 4a suggests….

L383 Same suggestion for

Similarly, linear systems (shelterbelts and windbreaks) appear to have a positive influence…

Author Response

We would like to thank the reviewer for their very helpful comments, which have greatly improved the manuscript. We have given point by point responses below:

Point 1: L79-80 What are the impacts of temperate agroforestry systems on sheep and cattle productivity, environmental impacts conditions, and farm economic viability?”.

Response 1: The term “environmental impacts” is used throughout the paper to refer to the negative environmental consequences of ruminant livestock production (first introduced on Lines 42-44). We chose this term in preference to “environmental conditions” as this instead implies factors such as temperature, precipitation and soil properties (i.e. conditions of the environment, rather than impacts on the environment).

Point 2: L106 ROSE …..first apparition add….RepOrting standards for Systematic Evidence Syntheses

Response 2: We have included the definition on first usage (Lines 110-111).

Point 3: Suggestion for a better link with Fig 4…

L358 Considering semi-natural systems (dehesa, montado, wood pasture and forest grazing systems) compared to pasture without trees, Figure 4a suggests….

L383 Same suggestion for

Similarly, linear systems (shelterbelts and windbreaks) appear to have a positive influence…

Response 3: We have amended as suggested, on Lines 393 and 422.

Reviewer 2 Report

under Outcomes – Economics, please, see: External economic benefits and social goods from prairie shelterbelts DOI 10.1007/s10457-008-9126-5

Line 14: what about increasing global population? It can be a factor as well.  

Table 1 can be improved: make it more compact by reducing empty spaces, for example. 

Line 106: please briefly describe ROSES 

Line 147: CEE stands for? 

Would be interesting inserting a topic to discuss directions farmers and policy makers should take towards a carbon-less production. What is the difference between cattle and sheep management to reduce C release and sequester more C? (Manure treatment, trees around the pasture…) What did you learned? A table may help 

Line 395: if the study is of no relevance, it does not increase the screening effort.  

Line 14: what about increasing global population? It can be a factor as well.  

Table 1 can be improved: make it more compact by reducing empty spaces, for example. 

Line 106: please briefly describe ROSES 

Line 147: CEE stands for? 

Would be interesting inserting a topic to discuss directions farmers and policy makers should take towards a carbon-less production. What is the difference between cattle and sheep management to reduce C release and sequester more C? (Manure treatment, trees around the pasture…) What did you learned? A table may help 

Line 395: if the study is of no relevance, it does not increase the screening effort.  

Author Response

We would like to thank the reviewer for their very helpful comments, which have greatly improved the manuscript. We have given point by point responses below:

Point 1: under Outcomes – Economics, please, see: External economic benefits and social goods from prairie shelterbelts DOI 10.1007/s10457-008-9126-5

Response 1: We have included this citation on Line 375 (citation number 84)

Point 2: Line 14: what about increasing global population? It can be a factor as well.

Response 2: We have incorporated the importance of this factor on Line 41.

Point 3: Table 1 can be improved: make it more compact by reducing empty spaces, for example. 

Response 3: We have changed Table 1 (Line 53) from portrait to landscape formatting, as had been done with Table 4. This has reduced the amount of empty space. The spaces between lines within cells is a feature of the journal’s table template and would have to be changed by the editor.

Point 4: Line 106: please briefly describe ROSES 

Response 4: We have inserted the definition of the acronym on Lines 110-111.

Point 5: Line 147: CEE stands for? 

Response 5: We have inserted the definition on Line 109.

Point 6: Would be interesting inserting a topic to discuss directions farmers and policy makers should take towards a carbon-less production. What is the difference between cattle and sheep management to reduce C release and sequester more C? (Manure treatment, trees around the pasture…) What did you learned? A table may help 

Response 6: We would like to thank the reviewer for raising this important and interesting point. We hope that our evidence map will be used to contribute to answers to exactly these sorts of questions. However, we don’t want to introduce a more speculative discussion of options for policy makers and practitioners into this paper, which is seeking to objectively present the evidence base without introducing bias. We have amended the manuscript to acknowledge this wider context on Lines 274-275, and cited some published reviews of mitigation options.

Point 7: Line 395: if the study is of no relevance, it does not increase the screening effort.  

Response 7: In order to determine whether a study is of relevance, citations returned from the searches of citation indexes and bibliographic databases were screened at title, abstract and then full text level (Lines 127-151). Even if a study does not meet the systematic map’s inclusion criteria, it may still have to be screened at full text level to determine this. We have amended Lines 436-439 to clarify our meaning on this.